# Neural Architecture Search Finds Robust Models by Knowledge Distillation

**Utkarsh Nath**[1]  **Yancheng Wang**[1]  **Yingzhen Yang**[1]

[1]School of Computing and Augmented Intelligence, Arizona State University, Tempe, AZ 85281, USA ,
{unath,ywan1053,yingzhen.yang}@asu.edu

## Abstract

Despite their superior performance, Deep Neural Networks (DNNs) are often vulnerable to adversarial attacks. Neural Architecture Search (NAS), a method for automatically designing the architectures of DNNs, has shown remarkable performance across various machine learning applications. However, the adversarial robustness of architectures learned by NAS against adversarial threats remains under-explored. By integrating a robust teacher, we examine whether NAS can yield a robust neural architecture by inheriting robustness from the teacher. In this paper, we propose Robust Neural Architecture Search by Cross-Layer Knowledge Distillation (RNAS-CL), a novel NAS algorithm that enhances the robustness of architectures learned by NAS through employing cross-layer knowledge distillation from a robust teacher. Distinct from previous knowledge distillation approaches that only align student-teacher outputs at the final layer, RNAS-CL dynamically searches for the optimal teacher layer to guide each student layer. Our experimental findings validate the effectiveness of RNAS-CL, demonstrating that it can generate both compact and adversarially robust neural architectures. Our results pave the way for developing new strategies for compact and robust neural architecture design applicable across various fields. The code of RNAS-CL is available at https://github.com/Statistical-Deep-Learning/RNAS-CL.

## 1 INTRODUCTION

Neural Architecture Search (NAS) has emerged as a vital tool for fostering advancements in deep neural networks, enhancing state-of-the-art (SOTA) performance across various fields, such as computer vision and natural language processing. NAS methods automate the search for neural architectures based on predefined criteria, eliminating the need for labor-intensive and time-consuming manual architecture design. Early works on NAS utilized Evolutionary Algorithms (EA) (Real et al., 2017) and Reinforcement Learning (RL) (Zoph and Le, 2017; Tan et al., 2019). Despite their effectiveness, these methods require substantial computational resources. For example, some of these approaches require up to thousands of GPU days to reach SOTA performance for the image classification task on the ImageNet dataset. To overcome these challenges, recent works (Liu et al., 2019; Cai et al., 2019; Wu et al., 2019; Wan et al., 2020; Nath et al., 2020) represent architectures with a shared-weight supernet and refine the weights through gradient descent. The architectures identified by the architecture parameters in the supernet through NAS deliver two key benefits. First, they are optimized for both speed and size, enhancing their practical utility. Second, the searched architectures set new SOTA performance for a variety of computer vision tasks. Both advantages make NAS incredibly useful for real-world applications. Nonetheless, most NAS methods focus primarily on optimizing accuracy, parameters, or FLOPs, and the performance of searched architectures under adversarial attacks remains underexplored, which is crucial for implementing secure and resilient machine learning systems. Few studies (Yue et al., 2022; Ning et al., 2020; Xie et al., 2023) have examined NAS with the aim of enhancing both adversarial robustness and efficiency. In this paper, we introduce RNAS-CL, a NAS methodology that concurrently optimizes for accuracy, latency, and defense against adversarial attacks, without the need for robust training.

Adversarial examples are created by altering the inputs, typically by introducing small, intricate disturbances into a clean image, causing the model to incorrectly classify it. It is well-known that almost all deep neural networks are vulnerable to these adversarial attacks (Szegedy et al., 2014). Consequently, assessing the resilience of models to

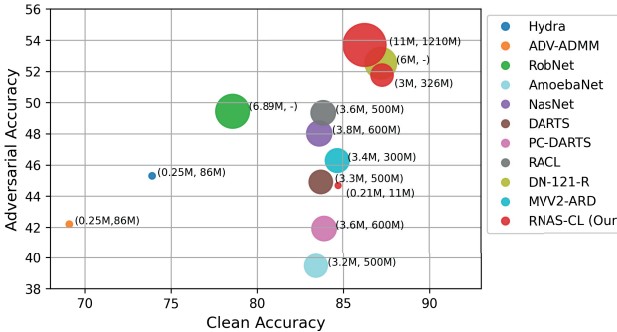

Figure 1: The figure compares various SOTA efficient and robust methods on CIFAR-10. Clean Accuracy represents top-1 accuracy on clean images. Adversarial Accuracy represents top-1 accuracy on images perturbed by PGD attack. Larger marker size indicates larger architecture. The numbers in brackets represent the number of parameters and MACs, respectively.

adversarial attacks is of paramount importance. Models that can withstand adversarial attacks are essential for critical applications such as autonomous driving, healthcare, and physical security systems.

Adversarial training is a well-established strategy to enhance the defense mechanisms of models against adversarial attacks (Goodfellow et al., 2015; Madry et al., 2018; Kannan et al., 2018; Tramèr et al., 2018; Zhang et al., 2019a). Approaches in this category usually train the models on adversarial examples, typically generated using techniques such as the fast gradient sign method (FGSM) (Goodfellow et al., 2015) or projected gradient descent (PGD) (Madry et al., 2018). Other defense strategies include training models with specialized loss functions or regularization (Cissé et al., 2017; Hein and Andriushchenko, 2017; Yan et al., 2018; Pang et al., 2020), preprocessing inputs prior to model input (Dziugaite et al., 2016; Guo et al., 2018; Xie et al., 2019), and employing model ensembles (Kurakin et al., 2018; Liu et al., 2018).

Recent studies have also highlighted the role of network architecture in influencing adversarial robustness (Madry et al., 2018; Guo et al., 2020; Su et al., 2018; Xie and Yuille, 2020; Huang et al., 2021). Inspired by these insights, we introduce Robust Knowledge Distillation for Neural Architecture Search (RNAS-CL). To the best of our knowledge, our work is among the first method that employs knowledge distilled from a robust teacher model to discover a robust architecture. Knowledge distillation traditionally involves transferring knowledge from a complex teacher model to a simpler student model using the teacher's outputs as "soft labels" (Hinton et al., 2015). However, beyond the outputs, the teacher's intermediate layers offer valuable attention information, where each layer focuses on different aspects

of the input (Zagoruyko and Komodakis, 2017).

The central question of our investigation is: *can a robust teacher improve the robustness of the student model by providing information about where to look, i.e., where to pay attention?* The proposed RNAS-CL method confirms this, enabling the student model to learn not only from the teacher's outputs but also "where to look" from the teacher's layers. Given the disparity in the number of layers between the teacher and student models, it is crucial for the student to identify the most beneficial teacher layer to learn from. The RNAS-CL method also involves searching for the ideal teacher layer for each student layer.

Furthermore, inspired by recent progress in self-supervised and semi-supervised learning that emphasizes consistency between predictions from various augmented views, we propose a novel Confidence-Aware Consistency loss or CAC loss. The CAC loss aims to maximize prediction consistency between adversarial and original views of inputs. CAC is compatible with various adversarial training methodologies, such as TRADES. The experimental results demonstrate that RNAS-CL significantly surpasses most existing models without adversarial training in robust accuracy on the CIFAR-10 dataset. Furthermore, applying CAC and TRADES to adversarially train RNAS-CL models significantly enhances their robustness. The effectiveness of RNAS-CL extends to promising results on the large-scale ImageNet dataset as well.

## 1.1 CONTRIBUTIONS

Our contributions are detailed as follows.

First, we propose RNAS-CL – a novel method for searching neural architectures that optimize the trade-off between robustness and prediction accuracy in a differentiable way. To the best of our knowledge, RNAS-CL is the first NAS approach that simultaneously optimizes for robustness and prediction accuracy without the necessity of robust training. By incorporating a penalty on model size and inference cost, the architectures derived through RNAS-CL are more compact than those from other NAS methods. We compare RNAS-CL against other models known for their computational efficiency and robustness (Sehwag et al., 2020; Ye et al., 2019; Gui et al., 2019; Goldblum et al., 2020; Dong et al., 2020; Huang et al., 2021). RNAS-CL models of comparable size demonstrate superior performance in both clean and PGD accuracy on the CIFAR-10 dataset.

Second, we extend the field of Knowledge Distillation (KD) within the framework of NAS. Unlike traditional KD, which relies on fixed connections between the teacher and student models, RNAS-CL innovates by introducing learnable connections between layers of the teacher and the student models. This advancement not only enhances the efficacy of KD but also provides insights into the development of

future adversarially robust NAS methods.

## 2 RELATED WORK

### 2.1 KNOWLEDGE DISTILLATION

Knowledge Distillation (KD) involves transferring knowledge from a larger, more complex model to a smaller, more manageable one. Hinton et al. (2015) introduced the concept of the teacher-student model, utilizing soft targets from the teacher to train the student model. This approach encourages the student to generalize in a manner similar to the teacher. Since this foundational work, various KD variants have been developed (Romero et al., 2015; Yim et al., 2017; Zagoruyko and Komodakis, 2017; Li et al., 2019; Tian et al., 2020a; Sun et al., 2019), incorporating feature maps, attention maps, or contrastive learning elements. FitNets (Romero et al., 2015) pioneered the use of intermediate-level hints from the teacher model to enhance student model training. This method involves a two-stage training process where the student first learns to predict the output of a middle (hint) layer of the teacher, followed by fine-tuning with the standard KD optimization function. The introduction of intermediate hints allowed the student model to achieve improved performance with fewer parameters. The introduction of intermediate hints allowed the student model to achieve improved performance with fewer parameters. Moving a step further, (Yim et al., 2017), (Zagoruyko and Komodakis, 2017) and (Li et al., 2019) utilize information from multiple teacher layers to guide the student's training. (Yim et al., 2017) utilized a Gramian matrix between the outputs of the first and last layers to chart the problem-solving process, transferring knowledge by minimizing the distance between the student's and teacher's flow matrices. (Li et al., 2019) calculated inter-layer and inter-class Gramian matrices to identify the most representative layers, minimizing the distance between these key layers of both student and teacher. (Zagoruyko and Komodakis, 2017) focused on minimizing the distance between the attention maps of the teacher and student at various blocks. In contrast with the above methods, RNAS-CL aims to map each student layer to a corresponding teacher layer, optimizing the match for each pair. This method extends the concept of attention map alignment, similar to that in (Zagoruyko and Komodakis, 2017), by minimizing the distance between the attention maps of matched student-teacher layers. This comprehensive mapping ensures a more detailed and effective knowledge transfer throughout the student's architecture.

### 2.2 NEURAL ARCHITECTURE SEARCH

Neural Architecture Search (NAS) is a method that automates the design of neural networks without human intervention. Traditionally, finding the optimal architecture within a given search space involves training each potential architecture from scratch until convergence. This approach, while straightforward, is computationally prohibitive. Early NAS efforts employed Reinforcement Learning (RL) (Zoph and Le, 2017; Tan et al., 2019) and Evolutionary Algorithms (EA) (Real et al., 2017), but these methods also demanded significant computational resources. More recent advancements (Liu et al., 2019; Cai et al., 2019; Wu et al., 2019) have introduced the concept of a weight-sharing super-network, which encompasses all candidate architectures. This network is over-parameterized and includes distinct paths for each architecture, each path having its own set of weights. These weights are then optimized through gradient descent during training to eventually select a single, optimal architecture. This selected network is subsequently trained in a conventional manner. While these methods have achieved state-of-the-art (SOTA) results on various classification tasks, their vulnerability to adversarial attacks remains largely unexplored. Research (Devaguptapu et al., 2021; Guo et al., 2020; Li et al., 2021; Madry et al., 2018; Su et al., 2018; Xie and Yuille, 2020; Huang et al., 2021) has shown that network architecture significantly influences adversarial robustness. Studies like (Devaguptapu et al., 2021) have noted that handcrafted architectures tend to be more resilient against adversarial attacks compared to NAS-generated models. Moreover, it has been empirically observed that larger models generally exhibit greater robustness against such attacks. (Guo et al., 2020) found that architectures with dense connections are particularly resistant to adversarial threats, prompting them to devise a NAS strategy that includes adversarial training on a supernet followed by the selection of densely connected architectures. (Li et al., 2021) expanded the backbone network to maintain accuracy while optimizing both the architecture and its parameters through adversarial training. Although this approach shows promising results, the main downside is that adversarial training is time-intensive and tends to degrade performance on standard (clean) images. Our proposed RNAS-CL method stands out by optimizing for both robustness and prediction accuracy without the need for adversarial training.

### 2.3 EFFICIENT AND ROBUST MODELS

The deep learning research community has thoroughly investigated the creation of efficient models and adversarially robust models as separate endeavors. However, integrating these two domains, that is, developing models that are both efficient and adversarially robust, has seen limited exploration. (Sehwag et al., 2020) introduced an approach to make pruning techniques sensitive to robust training objectives. They framed pruning as an empirical risk minimization (ERM) problem and combined it with a robust training framework. (Huang et al., 2021) examined how the configurations of network width and depth affect the robustness of

adversarially trained deep neural networks (DNNs). They found that reducing the capacity of the final blocks of a network could enhance its adversarial robustness. (Goldblum et al., 2020) developed Adversarially Robust Distillation (ARD), a method that prompts student networks to approximate their teacher's output within $\epsilon$-ball of training samples, fostering robustness in the student models. Additionally, a few Neural Architecture Search (NAS) methods (Yue et al., 2022; Ning et al., 2020; Xie et al., 2023) have aimed to optimize for accuracy, latency, and robustness concurrently. (Ning et al., 2020) implemented a multi-shot NAS approach to identify architectures that are robust against adversarial attacks, blending multiple one-shot methods to target specific capacities. (Xie et al., 2023; Yue et al., 2022) employed a one-shot NAS technique that selects an efficient model from an adversarially trained supernet. In comparison to these methods, models developed using RNAS-CL achieve superior accuracy on both clean and adversarial images while maintaining comparable size, thus demonstrating the effectiveness of integrating robustness and efficiency in neural architecture design.

# 3 ROBUST KNOWLEDGE DISTILLATION FOR NEURAL ARCHITECTURE SEARCH

We utilize knowledge distilled from a robust teacher model to facilitate the search for an architecture that achieves both robustness and efficiency. Knowledge distillation involves transferring knowledge from a larger teacher model to a smaller student model. In standard knowledge distillation, the teacher model's outputs serve as "soft labels" for training the student model. However, valuable attention information is also contained in the intermediate features of the teacher, where different layers concentrate on distinct parts of the input object. In RNAS-CL, the student model not only benefits from the teacher's soft labels but also learns where to direct its attention among the teacher's intermediate layers. Each student layer is specifically aligned with a robust teacher layer to learn targeted areas of focus. In Section 3.1, we discuss how we define attention maps. We hypothesize that learning directed attention from a robust teacher inherently enhances the student model's resistance to adversarial attacks. RNAS-CL is designed to identify the optimal tutor layer for each student layer while concurrently searching for an efficient architecture. In Section 3.2 and 3.3, we discuss our tutor and architecture search algorithm. Similar to other state-of-the-art NAS methods (Liu et al., 2019; Wu et al., 2019; Wan et al., 2020), RNAS-CL is structured around a search phase and a training phase. In the search phase, we optimize the neural architecture weights. In the training phase, the architecture selected in the search phase is trained using conventional methods. In Section 3.4, the objectives for the search and training phases are elaborated.

Although RNAS-CL can identify robust neural architectures for the student model, we aim to further enhance robustness through adversarial training. In Section 3.5, we introduce a novel regularization term, Confidence-Aware Adversarial Consistency Loss (CAC), which can be integrated with any adversarial training objective, such as TRADES and FastAT (Wong et al., 2020), to increase the robustness of the model.

## 3.1 ATTENTION MAP

We focus on learning where to pay attention from a robust teacher model, specifically analyzing convolution layers with activation tensors represented as $A \in R^{C \times H \times W}$ where $C$ is the number of channels, and $H$ and $W$ represent the spatial dimensions. A mapping function $\mathcal{F} : \mathbb{R}^{C \times H \times W} \longrightarrow \mathbb{R}^{H \times W}$ is defined to convert the tensor $A$ into an attention map $\mathcal{F}(A) \in \mathbb{R}^{H \times W}$ by $[\mathcal{F}(A)]_{hw} = \sum_{c=1}^{C} A_{c,h,w}^2$, where $A_{c,h,w}$ represents the element of $A$ with channel coordinate $c$ and spatial coordinates $h$ and $w$. This activation-based mapping function $\mathcal{F}$, which was introduced in (Zagoruyko and Komodakis, 2017), is applied post each convolution layer to generate an attention map. The mapping function $\mathcal{F}$ is applied to activation tensors after each convolution layer to generate an attention map. Several attention maps are illustrated in Figure 2(b). RNAS-CL aims to match each student layer with a corresponding teacher layer, termed as a tutor, ensuring that the student's attention map closely resembles that of its designated tutor from the teacher model. Given that the dimensions of the student's attention map might differ from that of its tutor, we interpolate all attention maps to a standardized dimension to facilitate accurate comparisons and alignments

## 3.2 TUTOR SEARCH

We aim to identify an appropriate tutor (teacher layer) for each student layer, which instructs on where to pay attention to, the potential combinations create a vast search space. Each student layer has the option to select any of the teacher layers as its tutor. Such flexibility results in a search space that grows exponentially with the number of layers in each model. For instance, the search space of a student model with 20 layers and a teacher model with 50 layers is of size $50^{20}$. To reduce the computation cost of the search process, we adopt Gumbel-Softmax (Jang et al., 2017) to search for the tutor for each student layer in a differentiable manner. Given network parameter $v = [v_1, \ldots, v_n]$ and a constant $\tau$. The Gumbel-Softmax function is defined as $g(v) = [g_1, \ldots, g_n]$ where $g_i = \frac{\exp[(v_i + \epsilon_i)/\tau]}{\sum_i \exp[(v_i + \epsilon_i)/\tau]}$ and $\epsilon_i \sim N(0, 1)$ is the uniform random noise, which is also referred to as Gumbel noise. When $\tau \to 0$, Gumbel-Softmax tends to the $\arg\max$ function. Gumbel-Softmax is a "reparametrization trick", that can be regarded as a differentiable approximation to the $\arg\max$ function.

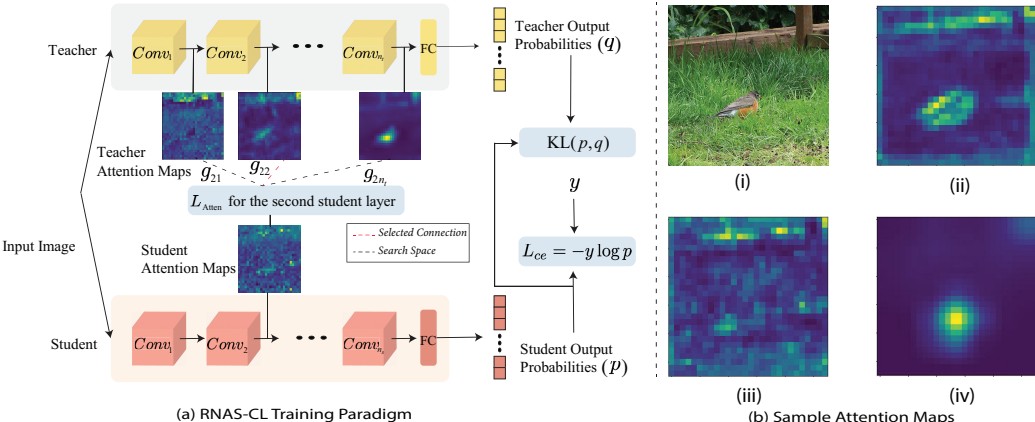

(a) RNAS-CL Training Paradigm

(b) Sample Attention Maps

Figure 2: (a) Training paradigm based on RNAS-CL. We connect attention maps from each student layer to each robust teacher layer. For each student layer, we search for the optimum teacher layer. $g_{ij}$ represents gumbel weights associated between $i^{th}$ student layer and $j^{th}$ teacher layer. RNAS-CL induces robustness to the student model by searching for the optimum teacher layer. We also search for the number of filters in each layer to build an efficient model inspired by FBNetV2 (Wan et al., 2020). (b) Sample attention maps corresponding to input Image (i) from low-level (ii), mid-level (iii), and high-level (iv) convolution layers.

Consider a teacher model $T$ and a student model $S$, each consisting of $n_t$ and $n_s$ layers respectively. Let $A_t^i$ and $A_s^i$ represent the activation tensors of the $i^{th}$ layer in the teacher and student models. In the RNAS-CL framework, each student layer ($i$) is paired with $n_t$ Gumbel weights ($g_i$), where $g_i$ belongs to the set $R^{1 \times n_t}$. Denote $g_{ij}$ as the Gumbel weight linking the $i^{th}$ student layer to the $j^{th}$ teacher layer. The attention loss is then defined as follows:

$$L_{\text{Attn}}(A_t, A_s) =$$
$$\frac{1}{n_s \times n_t} \sum_{i=0}^{n_s} \sum_{j=0}^{n_t} g_{ij} \left\| \frac{\mathcal{F}(A_s^i)}{||\mathcal{F}(A_s^i)||_2} - \frac{\mathcal{F}(A_t^j)}{||\mathcal{F}(A_t^j)||_2} \right\|_2, \quad (1)$$

where $A_s$ and $A_t$ represent the activation tensors for all convolution layers in the student and teacher models, respectively. $\mathcal{F}$ is the mapping function as defined in Section 3.1, $\| \cdot \|_2$ is the $\ell^2$-norm. Throughout the search process, we apply an exponential decay to the temperature $\tau$ of the Gumbel-Softmax, resulting in an encoding that closely approximates a one-hot vector.

### 3.3 ARCHITECTURE SEARCH

In addition to identifying the optimal tutor for each layer, we aim to develop an architecture that prioritizes efficiency and low latency. Drawing inspiration from FBNetV2 (Wan et al., 2020), our search focuses on determining the ideal number of filters, or output channels, for each convolution block. Consider a set of filter options $A = \{f_1, f_2, ..., f_n\}$ and their corresponding outputs $\{z_1, z_2, ..., z_n\}$ for a convolution block. The cumulative output is then defined as

$Z = \sum_{i=1}^{n} g_w^{(i)} z_i$, where $g_w^{(i)}$ represents the Gumbel weight associated with the $i^{th}$ filter choice. We optimize the number of FLOPs to achieve minimal latency, noting that FLOPs are directly proportional to the number of filters. This cumulative count of filters, influenced by the Gumbel weights, allows for differential optimization using SGD. Similar to the tutor search, the temperature decay is applied exponentially to secure an encoding nearing a one-hot vector. Figure 4 in the appendix illustrates the architecture search process by FBNetV2.

### 3.4 RNAS-CL LOSS

Adhering to the practices of leading NAS methodologies (Liu et al., 2019; Wu et al., 2019), RNAS-CL incorporates distinct searching and training phases. During the search phase, the Gumbel weights and other model parameters are updated each epoch. These include the Gumbel weights $\left\{ g_w^{(i)} \right\}$ associated with the student-teacher connections referenced in (1), and the Gumbel weights $\left\{ g_w^{(i)} \right\}$ for selecting filters as outlined in Section 3.3. Optimization of these weights is conducted using the RNAS-CL search loss, which will be detailed subsequently.

**RNAS-CL search loss.** Let $y$ be the ground-truth one-hot encoded vector, $p$ and $q$ be output probabilities of the student and teacher network, and $A_s$, $A_t$ as the activation tensors for all student and teacher convolution layers. The RNAS-CL

search loss is given by

$$L(y, p, q, A_t, A_s) = (-y \log p + \text{KL}(p, q) + \gamma_s L_{\text{Attn}}(A_t, A_s))n_f, \quad (2)$$

where $\text{KL}(p, q) = \sum_i p_i \log \frac{p_i}{q_i}$ denotes the Kullback-Leibler (KL) divergence between the probability distributions. $L_{\text{Attn}}$ is the attention loss as defined in (1) and $\gamma_s$ is a normalization constant. $n_f$ represents the latency, which is minimized through differential optimization as in (Wan et al., 2020).

Upon completion of the search phase, a tutor layer $j^*$ is chosen for each student layer $i$, where $j^* = \arg\max_j g_{ij}$. Additionally, the optimal filter choices for each convolution block, as discussed in Section 3.3, are determined based on the highest Gumbel weights. Subsequent to the search, the training phase commences, wherein the searched architecture is trained utilizing the RNAS-CL training loss, which will be delineated subsequently.

**RNAS-CL train loss.** Let $y$ be the ground-truth one-hot encoded vector, $p$ and $q$ be output probabilities of the student and teacher network, and $A_t, A_s$ be activation tensors for all student and teacher convolution layers. The training loss of RNAS-CL is

$$L(y, p, q, A_t, A_s) = L_{\text{CE}}(y, p) + \text{KL}(p, q) + \gamma_t L_{Attn}(A_t, A_s), \quad (3)$$

where $L_{\text{CE}}(y, p) = -y \log p$ represents the cross-entropy loss, $\text{KL}(p, q)$ denotes the KL-divergence, and $\gamma_t$ is a normalization constant. It should be noted that $g_i$ within $L_{\text{Attn}}$ is defined as a one-hot vector, leading to the optimization of each student attention map with respect to a specific tutor layer.

### 3.5 CONFIDENCE-AWARE ADVERSARIAL CONSISTENCY LOSS

Motivated by studies in self-supervised learning (Zhai et al., 2019) and semi-supervised learning (Berthelot et al., 2019) that emphasize the alignment of predictions across varied augmented views, we introduce a consistency loss aimed at enhancing the agreement between predictions from both adversarial and original views of the input data. This loss function is applied selectively to samples where the adversarial view yields highly confident predictions. For an input image $x$, its adversarial counterpart $x_{adv}$ is first generated, followed by acquiring the predictions for both $x$ and $x_{adv}$ from the student network, denoted as $p$ and $p_{adv}$, respectively. We then compute the mean of these predictions as $\bar{p} = \frac{p + p_{adv}}{2}$. Subsequently, this average prediction $\bar{p}$ is refined through the formula $\tilde{p}_j = \bar{p}_j^{\frac{1}{\tau}} / \sum k = 1^K \bar{p}_k^{\frac{1}{\tau}}$, where $K$ is the total number of classes and $\tilde{p}j$ is the $j$-th component of $\tilde{p}$. Here, $\tau \in (0, 1]$ serves as the sharpening

parameter. As $\tau$ decreases, $\tilde{p}$ approaches a one-hot distribution. This sharpened $\tilde{p}$ is then used as a pseudo label for $x$, reflecting the collective predictions from both $x$ and $xadv$. Our objective is to fortify the consistency between $p$ and $padv$ by minimizing their divergence from $\tilde{p}$, thereby defining our confidence-aware adversarial consistency loss as

$$L_{\text{CAC}}(x) = \mathbb{1}(\max(\bar{p}) \geq \gamma)(\text{KL}(\tilde{p}, p) + \text{KL}(\tilde{p}, p_{adv})), \quad (4)$$

where $\mathbb{1}(\cdot)$ represents the indicator function, and $\gamma \in [0, 1)$ denotes the confidence threshold. In the context of $L_{\text{CAC}}$, consistency optimization between the predictions of an image and its adversarial view is conducted only if the maximum value in the prediction vector $\bar{p}$ meets or exceeds $\gamma$. This condition ensures that $L_{\text{CAC}}$ enforces consistency solely on images where the predictions are deemed confident. The optimization of $L_{\text{CAC}}$ is designed to mitigate the detrimental effects of the noisy adversarial view, thereby enhancing the robustness of the student network. Our model undergoes adversarial training using $L_{\text{CAC}}$ alongside established adversarial objectives like TRADES and FastAT. The combined training loss for this adversarial training, incorporating both TRADES and $L_{\text{CAC}}$, is defined as

$$L_{\text{ADV}} = L_{\text{CAC}} + L_{\text{TRADES}} + L_{\text{KL}} + \gamma_t L_{\text{Attn}}, \quad (5)$$

where $L_{\text{TRADES}}$ is TRADES optimization objective and $L_{\text{KL}}, \gamma_t, L_{\text{Attn}}$ are the same as those in (3).

## 4 EXPERIMENTS

In this section, we present experiments conducted on real-world datasets to demonstrate the effectiveness of our proposed framework. The structure of this section is organized as follows. In Section 4.1, we discuss the settings and the implementation details. In Section 4.2, RNAS-CL is compared against state-of-the-art efficient and robust models on CIFAR-10, with more results outlined which are deferred to the supplementary.

### 4.1 IMPLEMENTATION DETAILS

In this paper, we assess the performance of RNAS-CL across three prominent public image classification benchmarks: (1) CIFAR-10, which includes $60k$ images distributed across 10 classes (Krizhevsky, 2009); (2) ImageNet, a comprehensive image classification dataset (Russakovsky et al., 2015) with approximately 1.2M images spanning 1000 classes; and (3) ImageNet-100, a more focused subset of the ImageNet-1k dataset (Russakovsky et al., 2015), featuring 100 classes and around $130k$ images (Tian et al., 2020c). We employ standard data augmentation techniques for each dataset, including random-resize cropping and random flipping. Initially, for each dataset, we undertake a searching step where our model is trained using the RNAS-CL search loss (2),

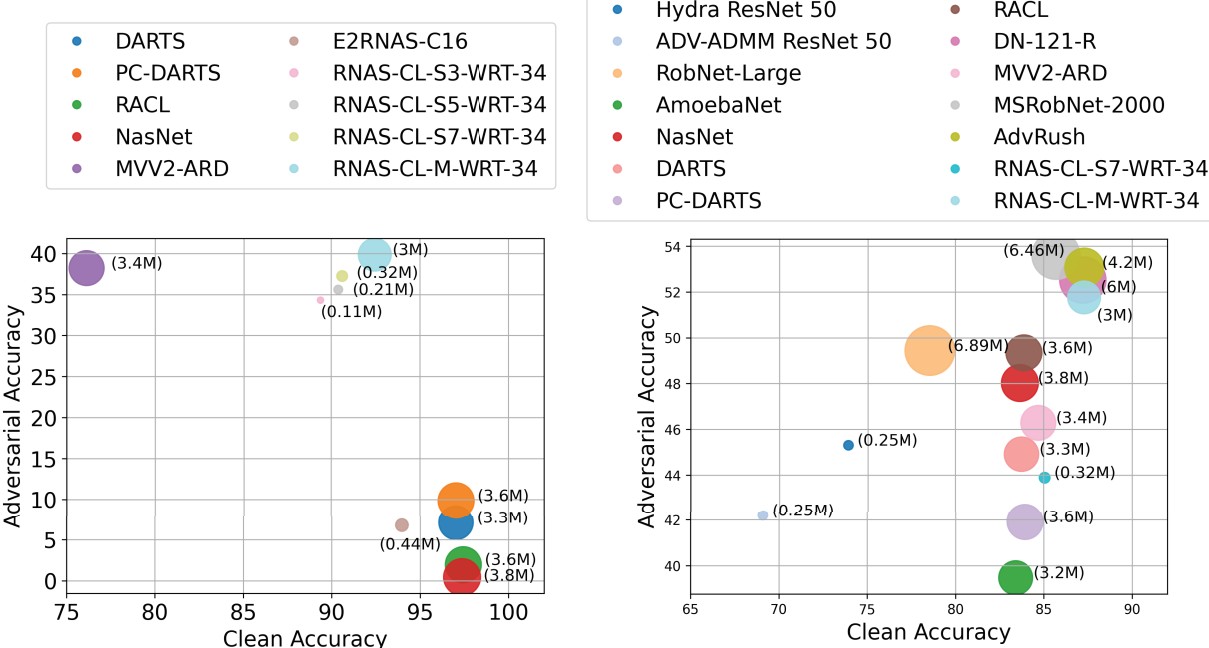

Figure 3: The figure compares the performance of various efficient and robust methods on CIFAR-10 dataset. Clean Accuracy represents top-1 accuracy on clean images. Adversarial Accuracy represents 20 step PGD attack.

aiming to identify optimal channel numbers and the appropriate connected teacher layers for each student layer. We explore various search spaces and utilize different robust teacher models throughout these experiments. In this paper, our model is denoted as RNAS-CL-X-T, where X indicates the search space and T denotes the robust teacher model. The search spaces are detailed in Table 2 and Table 3. We test four robust teacher models: ResNet-50, ResNet-18, WideResNet-50, and WideResNet-34, referred to as R-50, R-18, WRT-50, and WRT-34, respectively. For instance, RNAS-CL-S3-R-18 describes a model trained within the S3 search space using a ResNet-18 as the adversarially robust teacher model.

For all three datasets, we employ the SGD optimizer. The momentum and weight decay default values for ImageNet and ImageNet-100 are set at $0.9$ and $4 \times 10^{-5}$ respectively. The batch size used is $256$, and the learning rate starts at $0.1$, gradually decreasing to zero according to a cosine schedule. Post the search phase, which spans $100$ epochs, the identified architecture undergoes training from scratch for $200$ epochs using the RNAS-CL train loss (3). For CIFAR-10, the default momentum and weight decay values are $0.9$ and $2 \times 10^{-4}$ respectively, with a batch size of $128$. The model is trained over $100$ epochs in both the searching and training phases. The learning rate, initially set at $0.1$, is reduced by a factor of $10$ at the 75-th and 90-th epochs. In line with FBNetV2 settings, the Gumbel-Softmax tem-

perature ($\tau$) starts at $5.0$ and is exponentially reduced by $e^{-0.045}$ each epoch during the search phase. Hyperparameters $\lambda_s$ and $\lambda_t$ are maintained at $1.0$ across all experiments. During the search phase, $80\%$ of the data in each batch is used to optimize model weights while the remaining $20\%$ is employed for optimizing the architecture weights, the latter being Gumbel weights as discussed in Section 3.4. For robustness assessment, we deploy five prominent attacks: FGSM (Goodfellow et al., 2015), MI-FGSM (Dong et al., 2018), PGD (Madry et al., 2018), CW (Carlini and Wagner, 2017), and AutoAttack (Croce and Hein, 2020). Following standard practices in adversarial studies (Madry et al., 2018; Zhang et al., 2019b), adversarial perturbations are configured under the $\ell_\infty$ norm with a maximum perturbation limit of $8/255 (= 0.031)$.

## 4.2 EXPERIMENTAL RESULTS AND ABLATION STUDY

In this section, we evaluate the robustness of our method compared to other state-of-the-art (SOTA) efficient and robust models. These results are visually illustrated in Figure 3, where RNAS-CL models are positioned in the top right corner, indicating that they are among the models with the highest clean and adversarial accuracy. As shown in Figure 3, we benchmark RNAS-CL against both efficient models that have undergone adversarial training and those that have

not. All RNAS-CL models utilize a robust WideResNet-34 (Rice et al., 2020) as the teacher model. The results demonstrate that RNAS-CL substantially outperforms all models trained without adversarial interventions in terms of adversarial accuracy. Despite their smaller size, our RNAS-CL models achieve significantly better adversarial accuracy compared to their counterparts trained without adversarial measures. For instance, *RNAS-CL-S7-WRT-34* achieves more than a $28\%$ higher PGD accuracy than most other models of comparable size.

Next, we extend our comparison of RNAS-CL to models that have been adversarially trained. To ensure a fair assessment, after the initial training phase, our RNAS-CL models undergo further enhancement with our specialized adversarial training loss (5)for an additional 20 epochs. This phase of adversarial training boosts the adversarial accuracy of RNAS-CL models, allowing them to match or exceed the adversarial accuracy of other robustly trained models. Additionally, RNAS-CL models are significantly smaller and achieve considerably higher clean accuracy. For instance, the RNAS-CL-M-WRT-34 model not only matches but in some cases surpasses the adversarial accuracy of most other methods, while also being more compact and achieving significantly higher clean accuracy. Moreover, RNAS-CL enables the creation of notably smaller models. The Tiny RNAS-CL model, specifically RNAS-CL-S5-WRT-34, outperforms its counterpart, Hydra ResNet 34 (Sehwag et al., 2020), by over approximately $12\%$ in clean accuracy while maintaining the same model size.

**Comparison against various perturbation budgets.** To further highlight the efficacy of RNAS-CL, we contrast it with previously proposed defense mechanisms across a range of perturbation budgets. In Figure 6 of the supplementary, we illustrate the performance of various methods under PGD and FGSM attacks. For both types of attacks, RNAS-CL consistently outperforms its counterparts at every level of perturbation. Notably, as the size of the perturbation increases, the superiority of RNAS-CL becomes even more pronounced. Specifically, at a perturbation level of $\epsilon = 0.1$, RNAS-CL surpasses other methods by approximately $20\%$ in terms of resistance to both PGD and FGSM attacks. This robust performance underscores the strength of RNAS-CL in maintaining higher adversarial accuracy under increasingly challenging conditions.

We present additional experimental results and an ablation study in Section D of the supplementary. In Section D.1, our methods are benchmarked against various knowledge distillation techniques as detailed in (Park et al., 2019; Ahn et al., 2019; Tung and Mori, 2019; Tian et al., 2020b; Passalis and Tefas, 2018). Section D.2 evaluates RNAS-CL and the approach by (Huang et al., 2021) against recent attacks such as $CW_\infty$ (Carlini and Wagner, 2017) and AutoAttack (Croce and Hein, 2020) on the CIFAR-10 dataset. In Section D.3, we compare our model with the SOTA compact and efficient method (Huang et al., 2021), which is known for achieving one of the best PGD accuracies on ImageNet. Section D.4 provides ablation studies highlighting the significance of student-teacher cross-layer connections in RNAS-CL. We outline three training paradigms: the first uses standard cross-entropy loss without any teacher model, referred to as standard; the second minimizes the cross-entropy loss and standard KL Divergence with a robust teacher model, denoted as KL-X-T, where X represents the search space and T is the teacher model; the third model type, RNAS-CL, incorporates all three terms: cross-entropy loss, KL Divergence, and cross-layer student-teacher connections.

Moreover, in Section A of the supplementary, we report the robustness of adversarially trained teacher models used throughout the paper on the CIFAR-10 dataset in Table 1. In Section B and Section C, we discuss the architectures of various proposed supernets used in RNAS-CL for the CIFAR-10 dataset and outline the neural architecture search process based on FBNetV2.

# 5 CONCLUSIONS

In this paper, we propose Robust Neural Architecture Search by Cross-Layer Knowledge Distillation (RNAS-CL), a novel NAS algorithm that enhances the robustness of the student model through cross-layer knowledge distillation from a robust teacher. RNAS-CL optimizes neural architectures in a differentiable manner, aiming to balance robustness with clean accuracy, and can be employed with or without robust training. Our experiments demonstrate that compact models trained using RNAS-CL surpass those trained without robust measures in terms of adversarial robustness. Furthermore, incorporating adversarial training into RNAS-CL significantly boosts its adversarial resilience. Upon undergoing robust training, RNAS-CL models exhibit comparable adversarial robustness to those trained robustly from the outset, yet achieve superior clean accuracy. As a direction for future research, we plan to integrate robust training during the architecture search phase to further improve the robustness of the models.

# ACKNOWLEDGMENTS

This material is based upon work supported by the U.S. Department of Homeland Security under Grant Award Number 17STQAC00001-07-00. The views and conclusions contained in this document are those of the authors and should not be interpreted as necessarily representing the official policies, either expressed or implied, of the U.S. Department of Homeland Security. This work is also partially supported by the 2023 Mayo Clinic and Arizona State University Alliance for Health Care Collaborative Research Seed Grant Program.

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

# Neural Architecture Search Finds Robust Models by Knowledge Distillation
# (Supplementary Material)

**Utkarsh Nath**[1]                **Yancheng Wang**[1]                **Yingzhen Yang**[1]

[1]School of Computing and Augmented Intelligence, Arizona State University, Tempe, AZ 85281, USA ,
{unath,ywan1053,yingzhen.yang}@asu.edu

## A  ROBUST TEACHER MODELS

In this section, we report the robustness of adversarially trained teacher models used throughout the paper on the CIFAR-10 dataset in Table 1.

Table 1: Robustness results for various teacher models on the CIFAR-10 dataset.

| Method | Clean | PGD[20] |
|---|---|---|
| WRT-34 | 86.07 | 58.33 |
| ResNet 18 | 84.59 | 55.54 |
| ResNet 50 | 87.03 | 49.25 |

Table 2: The table describes the search space for CIFAR-10. Depth represents the depth of each stage. For example, 3-3-3 represents three convolution blocks in each stage. All search spaces have three stages. Stage 1, Stage 2, and Stage 3 represent the filter choices for the corresponding stages. For example, at stage 3 of RNAS-CL-S3, we search among 4 output channels, (64, 60, 56, 52), for each convolution block.

| Search Space | Depth | Stage 1 | Stage 2 | Stage 3 |
|---|---|---|---|---|
| RNAS-CL-S3 | 3-3-3 | 16, 12 | 32, 28, 24, 20 | 64, 60, 56, 52 |
| RNAS-CL-S5 | 5-5-5 | 16, 12 | 32, 28, 24, 20 | 64, 60, 56, 52 |
| RNAS-CL-S7 | 7-7-7 | 16, 12 | 32, 28, 24, 20 | 64, 60, 56, 52 |
| RNAS-CL-M | 9-7-1 | 80, 76 | 160, 156, 152, 148 | 128, 124, 120, 116 |
| RNAS-CL-L | 9-7-1 | 160, 156 | 320, 316, 312, 308 | 256, 252, 248, 244 |

## B  ARCHITECTURE

In this section, we discuss architectures for various proposed supernets used in RNAS-CL for the CIFAR-10 and ImageNet-100 datasets. Table 2 describes the supernets used for CIFAR-10. We use supernets with three blocks. Super-nets used for ImageNet-100 are described in Table 3. For ImageNet-100, the number of blocks varies from 3 to 5.

## C  ARCHITECTURE SEARCH BY FBNETV2

RNAS-CL builds both an efficient and adversarially robust deep learning model. In this work, we use the training paradigm of FBNetV2 to search for efficient models. In Figure 4, we illustrate the searching process for neural architecture at a single convolution layer. Each filter choice is attached with a Gumbel weight. These Gumbel weights are optimized to select an efficient model.

Table 3: The table describes the search space for ImageNet and ImageNet-100. Similar to Table 2, depth represents the depth of each stage. For ImageNet, we have up to 5 stages. Stage 1, Stage 2, Stage 3, Stage 4, and Stage 5 represent the filter choices for their respective stages. For example, in stage 1, we search among 4 output channel options, (28, 24, 20, 16), for each convolution block.

| Search Space | Depth | Stage 1 | Stage 2 | Stage 3 | Stage 4 | Stage 5 |
|---|---|---|---|---|---|---|
| RNAS-CL-IS | 3-3-3 | 28, 24, 20, 16 | 40, 36, 32, 28 | 96, 88, 80, 72, 64, 56, 48 | | |
| RNAS-CL-IM | 3-3-3-4 | 28, 24, 20, 16 | 40, 36, 32, 28 | 96, 88, 80, 72, 64, 56, 48 | 128 120, 108, 100, 92, 84, 76, 68 | |
| RNAS-CL-I | 3-3-3-4-4 | 28, 24, 20, 16 | 40, 36, 32, 28 | 96, 88, 80, 72, 64, 56, 48 | 128 120, 108, 100, 92, 84, 76, 68 | 216, 208, 200, 192, 184,176, 168, 160, 152, 144,136, 128, 120, 108 |
| RNAS-CL-IL | 1-2-2-4-3 | 28, 24, 20, 16 | 40, 36, 32, 28 | 96, 88, 80, 72, 64, 56, 48 | 128 120, 108, 100, 92, 84, 76, 68 | 216, 208, 200, 192, 184,176, 168, 160, 152, 144,136, 128, 120, 108 |

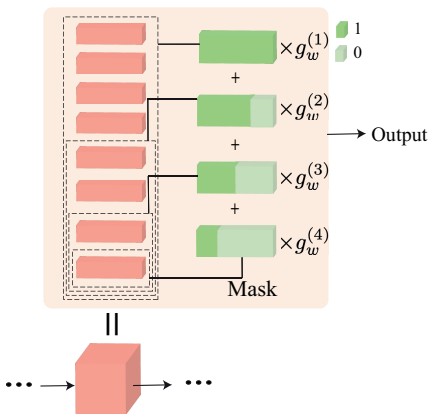

Figure 4: Illustration of searching for the neural architecture of each layer of student model using the searching mechanism in FBNetV2. $\left\{ g_w^{(i)} \right\}$ represents gumbel weights associated with different filter choices.

# D  MORE EXPERIMENTAL RESULTS

## D.1  COMPARISON AGAINST KD VARIANTS

In this section, we evaluate our methods in comparison to a variety of knowledge distillation (KD) techniques as outlined in (Park et al., 2019; Ahn et al., 2019; Tung and Mori, 2019; Tian et al., 2020b; Passalis and Tefas, 2018). We utilize Robust WRT-34 as the teacher model across all KD methods and train three distinct student architectures: RNAS-CL-S3, RNAS-CL-S5, and RNAS-CL-S7. In Figure 5, models trained under our paradigm are clearly positioned in the upper right-most part of the graph, underscoring the effectiveness of our intermediate cross-connections strategy. The RNAS-CL-S3 architecture, when trained using Relational Knowledge Distillation (RKD), demonstrates performance comparable to that achieved through our method. Beyond this, all models trained using the RNAS-CL approach significantly surpass other methods in both clean and adversarial accuracy, highlighting the robustness and efficiency of our training strategy.

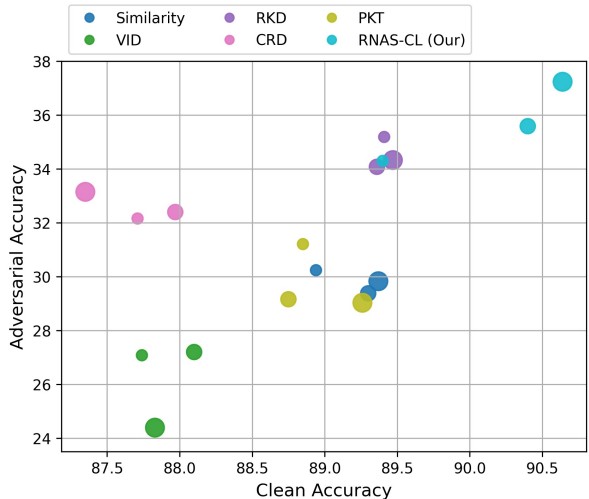

Figure 5: The figure compares various knowledge distillation variants (Similarity (Tung and Mori, 2019), VID (Ahn et al., 2019), RKD (Park et al., 2019), CRD (Tian et al., 2020b), PKD (Passalis and Tefas, 2018)) against RNAS-CL on the CIFAR-10 dataset. Adversarial Accuracy represents top-1 Accuracy on images perturbed by 20 step PGD attack. Clean Accuracy represents top-1 Accuracy on clean images. Larger marker size indicates larger architecture. For each method, RNAS-CL-S3, RNAS-CL-S5, and RNAS-CL-S7 are represented by increasing marker size.

## D.2 COMPARE CIFAR-10 MODEL AGAINST CW AND AUTOATTACK

In this section, we evaluate the performance of RNAS-CL and the approach described by (Huang et al., 2021) against recent adversarial attacks, specifically $CW\infty$ (Carlini and Wagner, 2017) and AutoAttack (Croce and Hein, 2020), using the CIFAR-10 dataset. The $CW\infty$ attacks, originally designed to overcome defensive distillation, are implemented here in their $\ell_\infty$ variant, optimized using PGD with a maximum perturbation budget of $\epsilon = 8/255$. AutoAttack, known for being a parameter-free ensemble attack, is currently regarded as one of the most robust and reliable benchmarks for evaluating adversarial defenses. The comparative results are shown in Table 4, showcasing how each model withstands these rigorous testing methods.

Table 4: Comparison between the performance of (Huang et al., 2021) and RNAS-CL against $CW_\infty$ (Carlini and Wagner, 2017) and AutoAttack (Croce and Hein, 2020) on the CIFAR-10 dataset.

| Method | $CW_\infty$ | AA |
|---|---|---|
| VGG-R (Huang et al., 2021) | 46.49 | 38.44 |
| DN-121-R (Huang et al., 2021) | 53.07 | 47.75 |
| RNAS-CL-S3-WRT-34 (Our) | 47.07 | 37.17 |
| RNAS-CL-S5-WRT-34 (Our) | 48.33 | 39.28 |
| RNAS-CL-S7-WRT-34 (Our) | 47.91 | 38.36 |
| RNAS-CL-M-WRT-34 (Our) | **53.52** | 46.89 |
| RNAS-CL-L-WRT-34 (Our) | 52.63 | **48.49** |

## D.3 RESULTS FOR IMAGENET

In this section, we compare our model against the SOTA compact and efficient method (Huang et al., 2021), which is known to achieve one of the best PGD accuracies using a compact and efficient model on ImageNet. In Table 5, we evaluate RNAS-CL and (Huang et al., 2021) against 10 step PGD attack with $\epsilon = 4/255$ on the ImageNet dataset. Both models are adversarially trained using FastAT (Wong et al., 2020). Next, we train RNAS-CL with FastAT and CAC to further increase the robustness. RNAS-CL models significantly outperform (Huang et al., 2021) in all three attributes: clean accuracy, robust accuracy, and the number of parameters.

Table 5: Performance of various efficient and robust methods on the ImageNet dataset. Clean and PGD are the same as that in Figure 3. ∗ represents approximate values.

| Method | Objective | Clean | PGD$^{10}$ | Params (M) | GFLOPs |
|---|---|---|---|---|---|
| ResNet-50-R (Huang et al., 2021) | FastAT | 56.63 | 31.14 | 25.5 | 4∗ |
| RNAS-CL-IL-WRT-50 | FastAT | 61.7 | 32.5 | 8.5 | 0.35 |
| RNAS-CL-IL-WRT-50 | FastAT + CAC | 61.5 | 33.5 | 8.5 | 0.35 |

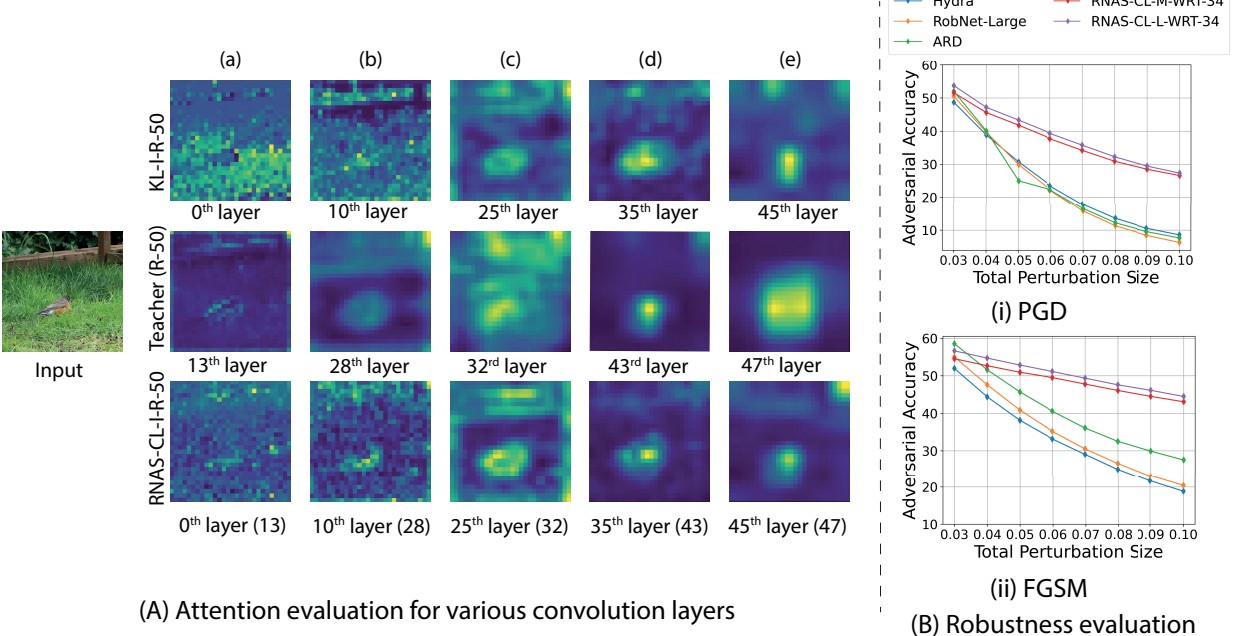

(A) Attention evaluation for various convolution layers

(B) Robustness evaluation

Figure 6: (A) KL-I-R-50 represents attention maps from a model trained using cross-entropy loss and knowledge distillation without any cross-layer connections. Teacher and RNAS-CL represent attention maps from the robust teacher (ResNet-50) and RNAS-CL model. The name for each RNAS-CL layer includes its connected teacher layer. For example, in the 0-th layer (13), 13 represents the corresponding teacher layer. RNAS-CL drives attention maps from student layers closer to their corresponding teacher layers. (B) Robustness evaluation under different perturbation sizes for PGD and FGSM attacks on CIFAR-10.

## D.4 ABLATION STUDY

Table 6: Ablation study on various components used during RNAS-CL training on the CIFAR-10 dataset with RNAS-CL-S7-WRT-34 as the base model. CE represents models trained using Cross-Entropy Loss. CE + KL represents models trained by minimizing the Cross-Entropy loss and standard KL Divergence with a robust teacher model. CE + ICC represents models trained by minimizing the Cross-Entropy loss and Intermediate Cross-Connections (ICC). Clean and PGD are the same as that in Figure 3.

| Training Type | Objective Function | Clean | PGD$^{20}$ |
|---|---|---|---|
| Without Adversarial training | CE | 90.98 | 19.3 |
| | CE + KL | 90.76 | 36.3 |
| | CE + ICC | 90.33 | 35.54 |
| | CE + KL + ICC | 90.62 | 37.24 |
| With Adversarial training | CE | 80.85 | 39.67 |
| | CE + KL | 85.07 | 41.63 |
| | CE + ICC | 82.45 | 41.03 |
| | CE + KL + ICC | 85.06 | 43.88 |

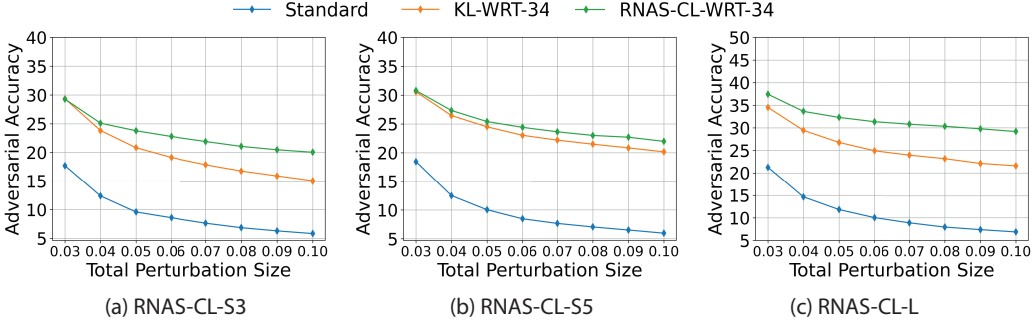

Figure 7: Adversarial accuracy of various models at various perturbation budgets on CIFAR-10.

This ablation study demonstrates the significance of student-teacher cross-layer connections in RNAS-CL. We compare three types of training paradigms. In the first training paradigm, we conduct searching and training using cross-entropy loss without any teacher model. We refer to this as standard. In the second paradigm, we conduct searching and training by minimizing the cross-entropy loss and standard KL Divergence with a robust teacher model. We refer to the corresponding models as KL-X-T, where X represents the search space, and T represents the robust teacher model. Finally, the third model type is RNAS-CL, where we include all three terms, cross-entropy loss, KL Divergence, and cross-layer student-teacher connections.

In Figure 6(A), we compare the attention maps from student models trained using RNAS-CL-I-R-50 against students trained using KL-I-R-50. We compare attention maps for various convolution layers at regular intervals. As expected, adding cross-layer connections obtains attention maps from the student model closer to the teacher model. Each student layer learns where to pay attention from its connected teacher layer. For example, in column (b), the KL-I-R-50 layer attends to various parts of the image, whereas the RNAS-CL layer learning from the 28-th teacher layer pays more attention to the informative central part of the image. Similarly, in column (c), the RNAS-CL layer learns from the teacher model to pay more attention to the central and upper portions of the image. In Table 6, we compare the performance of various components of RNAS-CL. We observe that under both training schemes, KL and ICC (Intermediate Cross-Connections) significantly increase the robustness compared to the standard network. Finally, combining KL and ICC, that is, RNAS-CL, outperforms its counterparts. In Figure 7, we compare RNAS-CL models against KL-X-T and standard models against PGD attacks at various perturbation budgets on the CIFAR-10 dataset.