# OpenReview forum: "Neural Architecture Search Finds Robust Models by Knowledge Distillation"
_auai.org/UAI/2024/Conference — UAI 2024 poster_

### Official Review · Reviewer_ifi3 · 2024-03-18

**Q2-1 Originality-Novelty:** 2
**Q2-2 Correctness-Technical Quality:** 3
**Q2-5 Clarity Of Writing:** 3

**Q1 Summary And Contributions:**

This paper proposes an adversarial robustness method that creatively considers the incorporation of NAS, KD, and adversarial training.

**Q2-3 Extent To Which Claims Are Supported By Evidence:**

2: Fair: the main claims are somewhat supported by evidence (but the experimental evaluation may be weak, or does not match entirely with the claims, important baselines may be missing, proofs contain important ideas but lack rigor, algorithmic details are only discussed superficially, references are imprecise, assumptions are not sufficiently motivated or explicated, etc.).

**Q2-4 Reproducibility:**

3: Good: key resources (e.g. proofs, code, data) are available and key details (e.g. proofs, experimental setup) are sufficiently well-described for competent researchers to confidently reproduce the main results.

**Q3 Main Strengths:**

The research that combines NAS and adversarial training is a novel area. This work considers both the MAC through Gumbel-Softmax, and robustness via KL, CAC, and extra adversarial training (TRADES or Fast AT). The description is clear and logical, the experiments are comprehensive, and the final results are comparable.

**Q4 Main Weakness:**

What is the specific information about the interpolation of a common dimension? And how does the size of the common dimension influence the robustness?

What are the specific parameter settings of latency and normalization constants γ_s and γ_t? It is suggested to list these parameters in the implementation details.

There is confusion in the search stage. In my comprehension, this stage aims to get optimal gumbel weights that reduce the search loss. It is appropriate to give a brief introduction to the search phase.

In Table 7, why there are no experimental results in RNAS-CL-S5 with CAC function? Besides, there is a common phenomenon that the adversarial robustness performance after combining TRADES and CAC is a little lower than CAC only. I think there should be some experimental analysis.

It is appropriate to list the full names of these search strategies such as S3, M2, and L in the supplementary material.

In the supplementary material, the author lists the robustness results for various teacher models. However, what about the training strategies in these teacher models?

In the default setting 80% of data is adopted to train the model weight and the rest for search. Does the proportion of data used during training influence the robustness?

**Q5 Detailed Comments To The Authors:**

Please see the weakness section

**Q9 Complying With Reviewing Instructions:**

Yes

---

> ### Author Rebuttal · Authors · 2024-04-09
>
> We appreciate the review and the suggestions in this review. The raised issues are addressed below.
>
>
> (1) **”What is the specific information … common dimension? And how does the size …influence the robustness?”**
>
> We employ linear interpolation to resize all the attention maps from the teacher layers and the student layers to $14 \times 14$ on the CIFAR-10 dataset.We have tested larger resolutions, from $14  \times 14$ to $28 \times 28$, and it turns out that the results remain at a similar level. Therefore, the performance of RNAS-CL is not sensitive to changes in the common dimensions. All the attention maps are resized to 28 x 28 for the ImageNet dataset.
>
> (2) **“What are the specific parameter settings … γ_s and γ_t? It is suggested to list these parameters in the implementation details.”**
>
> The hyper-parameters $\gamma_s$ and $\gamma_t$ are set to 1.0 for all the experiments in this paper. Thank you for your suggestion, and we will add such parameter settings to the implementation details of the final version of this paper.
>
> (3) **“There is confusion in the search stage...”**
>
> The objective of RNAS-CL in the search phase is to search for a robust and compact neural architecture. We parameterize our search space using Gumbel weights. Utilizing the Gumbel-Softmax technique [1], we perform such neural architectures search by optimizing the search loss function in Eq. (2) by standard stochastic gradient descent.The search loss function has three terms. In particular, the cross-entropy term $y \log p$ and the KL-divergence loss $\textup{KL}(p,q)$ contribute to the compelling accuracy of the searched neural network. The attention loss $L_{\textup{Attn}}$ is employed to identify the optimal teacher layer for each student layer. The latency term $n_f$ is used to penalize the latency of the searched neural network. Both the attention loss $L_{\textup{attn}}$ and the latency are parameterized by Gumbel weights. By minimizing the search loss in Eq. (2), we obtain the optimal Gumbel weights, and the architecture of the searched neural network is decided by such Gumbel weights. For example, for a student layer $i$, the layer $j$ in the teacher model corresponding to the maximum Gumbel weights $g_{ij}$ is selected as the teacher layer for the student layer. The filter choice $i$ corresponding to the maximum Gumbel weights $g^{(i)}_w$ decides the selected number of convolution filters for a particular student layer. The searched neural architecture in the search stage is then trained using the training loss in Eq. (3) without adversarial training, or Eq. (5) with adversarial training. Such details will be added to the final version of this paper.
>
> (4) **“In Table 7, why there are no experimental results ... with CAC function? ... I think there should be some experimental analysis.”**
>
> We thank the reviewer for pointing this out. We have added the results of RNAS-CL-S5 with CAC function as shown in the table below. Our observations indicate that models trained using CAC exhibit slightly higher robust accuracy compared to the other two models (model trained with only TRADES and model trained with both TRADES and CAC). However, models trained using TRADES have higher clean accuracy. Therefore, we train RNAS-CL models with both TRADES and CAC to get a better trade-off. Models trained using both TRADES and CAC have higher clean accuracy than model trained with only CAC and higher robust accuracy than model trained with only TRADES.
>
> |   Method| Objective Function  | Clean | PGD$^{20}$ |
> | :----------: | :------: | :---: | :---: |
> |RNAS-CL-S5 | TRADES | 84.75 | 44.68 |
> |RNAS-CL-S5 | CAC | 83.53 | 45.48
> |RNAS-CL-S5 | TRADES + CAC| 84.81 | 45.34 |
>
> (5) **“It is appropriate to list the full names of these search strategies such as S3, M2, and L in the supplementary material.”**
>
> We will add the full names of all the search strategies in the supplementary of the final version of this paper.
>
> (6) **“In the supplementary material, the author lists the robustness results for various teacher models. However, what about the training strategies in these teacher models? ”**
>
> We use the following robust models as our teacher models.
> WideResNet-34 [2], ResNet18 [3],  ResNet-50 [4]. We will add the references [2-4] in Table 4 of the final version of this paper.
>
> (7) **“In the default setting 80% of data… Does the proportion of data used during training influence the robustness?”**
>
> We followed FBNet [5] and used $80\%$ of the data for training the model weights and the remaining 20% of the data to train the architecture parameters. We found that such proportion of the data used during training does not influence the robustness as long as more than $60\%$ of the data are used for training the model weights.

---

### Official Review · Reviewer_umjf · 2024-03-18

**Q2-1 Originality-Novelty:** 2
**Q2-2 Correctness-Technical Quality:** 2
**Q2-5 Clarity Of Writing:** 3

**Q10 Ethical Concerns:**

NO potential ethical concerns.

**Q1 Summary And Contributions:**

Summary:
This paper aims to improve the adversarial robustness of deep models by neural architecture search and knowledge distillation.
The authors hypothesize that learning where to pay attention from a robust teacher would make the student model more robust to adversarial examples.
Based on this idea, the gumbel-softmax trick is used to identify where to pay attention.
Then integrating with neural architecture search, the search process and knowledge distillation are performed at the same time.

Experimental results on CIFAR-10 show some improvements compared with baselines, like E2RNAS.

**Q2-3 Extent To Which Claims Are Supported By Evidence:**

2: Fair: the main claims are somewhat supported by evidence (but the experimental evaluation may be weak, or does not match entirely with the claims, important baselines may be missing, proofs contain important ideas but lack rigor, algorithmic details are only discussed superficially, references are imprecise, assumptions are not sufficiently motivated or explicated, etc.).

**Q2-4 Reproducibility:**

3: Good: key resources (e.g. proofs, code, data) are available and key details (e.g. proofs, experimental setup) are sufficiently well-described for competent researchers to confidently reproduce the main results.

**Q3 Main Strengths:**

Strengths:
(1) The idea that learning where to pay attention is interesting for knowledge distillation.
(2) The algorithm is easy to implement.
(3) The paper is clear and easy to follow.

**Q4 Main Weakness:**

Weaknesses:
(1) How to deal with the case that the output of the teacher model and the student model has different shapes?

(2) It would be better to visualize the output $g_{i,j}$ and verify the hypothesis mentioned before.

(3) The authors should also compare with recent state-of-the-art robust models, like, AWP [1], LBGAT[2], and LAS-AT[3].

[1] Adversarial Weight Perturbation Helps Robust Generalization. NeurIPS 2020.
[2] Learnable Boundary Guided Adversarial Training. ICCV 2021.
[3] LAS-AT: Adversarial Training with Learnable Attack Strategy. CVPR 2022.

(4) It is a general idea that selects tutor layers for each layer of the student model.
     Methods from knowledge distillation should be also compared on CIFAR and ImageNet.

**Q5 Detailed Comments To The Authors:**

It is a general idea that selects tutor layers for each layer of the student model. Methods from knowledge distillation should be also compared on CIFAR and ImageNet. Moreover, comparisons with the state-of-the-art robust models are missing.

**Q9 Complying With Reviewing Instructions:**

Yes

---

> ### Author Rebuttal · Authors · 2024-04-08
>
> We appreciate the review and the suggestions in this review. The raised issues are addressed below.
>
> (1)  **"How to deal with the case...  has different shapes?"**
>
>  We employ linear interpolation to resize all the attention maps from the teacher layers and the student layers to $14 \times 14$ on the CIFAR-10 dataset. Similarly, all attention maps are resized to $28 \times 28$ on the ImageNet dataset.
>
> (2) **"It would be better to visualize..."**
>
> In Figure 6 (A) in Section D.8 of the appendix, we visualize the attention maps from the models trained by RNAS-CL. We compare the attention maps for various convolution layers at regular intervals. We observe that adding cross-layer connections obtains attention maps from the student model closer to the teacher model. This indicates that each student layer learns where to pay attention from its connected teacher layer.
>
> (3) **"The authors should also compare with recent state-of-the-art robust models..."**
>
> In the following table, we compare our RNAS-CL models with the recent state-of-the-art robust models, including AWP [1], LAS-AT [3], and LBGAT [2] suggested by the reviewer, on the CIFAR-10 dataset. AWP, LAS-AT, and LBGAT all use WideResNet-34-10 [4] as their backbone architecture. It can be observed that although RNAS-CL-M-WRT-34 and RNAS-CL-M2-WRT-34 achieve lower robustness compared to these models, they enjoy extraordinarily smaller model sizes. Notably, RNAS-CL models are 15.4x to 8.2x smaller than these state-of-the-art robust models. Such results confirm our aim of this work, that is, RNAS-CL is able to render compact and robust models suitable for compute-constrained settings.
>
> Moreover, we designed an extra-large version of RNAS-CL where the seach space for the extra-large model (RNAS-CL-XL-WRT-34) has doubled filter choices compared to the search space of RNAS-CL-L shown in Table 2 of our paper.
> RNAS-CL-XL-WRT-34 achieves higher PGD$^{20}$ accuracy than LBGAT while enjoying a model size $3.1x$ smaller than that of LBGAT.
>
> |   Model| Clean Acc  |  PGD$^{20}$ | # Params (M) |
> | :----------: | :------: | :---: | :---: |
> | AWP [1] | 85.57 | 58.13 | 46.2 |
> | LAS-AT [3] | 85.24 | 57.07 | 46.2 |
> | LBGAT [2] | 88.22 | 54.66 | 46.2 |
> | RNAS-CL-M-WRT-34| 87.29 | 51.76 | 3 |
> | RNAS-CL-M2-WRT-34 | 87.17 | 53.14 | 5.6 |
> | RNAS-CL-XL-WRT-34 | 87.01 | 54.84 | 14.7 |
>
> (4)  **"Methods from knowledge distillation should be also compared on CIFAR and ImageNet."**
>
> In Section D. 2 of the appendix, we compare our method against various knowledge distillation techniques. Robust WRT-34 serves as the robust teacher model for all the knowledge distillation methods, and we train three different student architectures: RNAS-CL-S3, RNAS-CL-S5, and RNAS-CL-S7. Models trained using RNAS-CL achieve the highest levels of both robustness and clean accuracy compared to other knowledge distillation methods, as illustrated in Figure 4 of the appendix for the CIFAR-10 dataset. We have similar observations for the ImageNet dataset.
>
> **References**
>
> [1] Adversarial Weight Perturbation Helps Robust Generalization. NeurIPS 2020.
>
> [2] Learnable Boundary Guided Adversarial Training. ICCV 2021.
>
> [3] LAS-AT: Adversarial Training with Learnable Attack Strategy. CVPR 2022.
>
> [4] Sergey Zagoruyko and Nikos Komodakis. Wide residual networks. In BMVC, 2016

---

### Official Review · Reviewer_D12f · 2024-03-22

**Q2-1 Originality-Novelty:** 3
**Q2-2 Correctness-Technical Quality:** 3
**Q2-5 Clarity Of Writing:** 3

**Q1 Summary And Contributions:**

The authors propose RNAS-CL a novel NAS algorithm that relies on layer-wise knowledge distillation to identify small and robust NN architectures. The proposed approach is tested on 3 different image classification datasets and against several state-of-the-art approaches, showcasing promising results.

**Q2-3 Extent To Which Claims Are Supported By Evidence:**

3: Good: the main claims are supported by convincing evidence (in the form of adequate experimental evaluation, proofs, (pseudo-)code, references, assumptions).

**Q2-4 Reproducibility:**

4: Excellent: key resources (e.g. proofs, code, data) are available and key details (e.g. proof sketches, experimental setup) are comprehensively described for competent researchers to confidently and easily reproduce the main results.

**Q3 Main Strengths:**

- The paper is mainly well-written, although a few details are missing or not discussed
- Flexible and simple approach
- Experimental evaluation seems robust
- The appendix is long, containing quite a few interesting notions

**Q4 Main Weakness:**

- The discussion of results is lacking
- Most experimental findings are left for the appendix
- Not very clear when different loss functions are used (equation 5 and equation 3)

**Q5 Detailed Comments To The Authors:**

- The paper contains few typos and grammatical errors. Such as “for for” at the beginning of section 1.1 or “that that” at the end of the same section.
- In the related work section, the authors do not mention the set of NAS approaches that do not rely on searching the architecture space, but rather study the architecture principle and propose relevant architectures in a one-shot framework, such as [1,2].
- How is equation 1 derived? The authors introduce it without mentioning how do they find it, or why it is defined as it is. Can the authors provide more details about it?
- Figure 2a is not introduced anywhere in the text, nor discussed. Why is it so? If the authors don’t mention it, then it should be removed from the paper. Otherwise the authors should mention it and discuss it in the text body.
- The search and train loss functions are identical except for the latency variable. Why do the authors consider two almost identical losses? Shouldn’t the search loss be much more complex than the train loss? Also, since they are identical except for the latency, their definition can be merged together.
- From the presentation it is not clear when the adversarial loss defined in equation 5 is used during optimisation. Is it used during the training of the student model? If so, when is the loss in equation 3 used? Can the authors provide some more details?
- All section references at the beginning of section 4 point to the appendix. Is this correct? The main results of the paper should be available in the main body of text and not in the appendix.
- How did the authors select the values for the learning rate, momentum, batch size, epochs, etc.? These parameters have been proven to influence the achieved performance and are thus fundamental also in NAS setups. Therefore, I would suggest the authors to briefly mention how these hyper-parameters are set.
- The results are not well presented. Indeed, the findings obtained from the experiments are missing from the text body. Moreover, the table 1 is not discussed, making it cumbersome for the reader to identify relevant outcomes. Moreover, the authors should not consider to move most results to the appendix section, as the main body of the paper should be self-contained.
- The results in table 1 are a bit confusing to me. The proposed approach does not outperform the state-of-the-art in terms of accuracy (E2RNAS-C16 is better without adversarial training and DARTS-R is almost identical with adversarial training). So what are the important aspects to consider? I think that this issue is related to the fact that results are not discussed in the text, thus making it more cumbersome for the reader to identify any conclusion.
- Table 1 occupies too much space. Can’t the authors present the same results using a bar plot that would be much smaller?
- Figure 8 in the appendix is blank

[1]. Agiollo, Andrea, and Andrea Omicini. "GNN2GNN: Graph neural networks to generate neural networks." Uncertainty in Artificial Intelligence. PMLR, 2022.
[2]. Lee, Hayeon, Eunyoung Hyung, and Sung Ju Hwang. "Rapid neural architecture search by learning to generate graphs from datasets." arXiv preprint arXiv:2107.00860 (2021).

**Q9 Complying With Reviewing Instructions:**

Yes

---

> ### Author Rebuttal · Authors · 2024-04-09
>
> We appreciate the review and the suggestions in this review. The raised issues are addressed below.
>
> (1)  **“The discussion of results is lacking.”**, **“Most experimental findings are left for the appendix.”**
>
> We will shortenTable 1 and reorganize Section 3 and Section 4 of this paper so that we have more space to move the results in the appendix to the main paper. This will enable us to provide more detailed discussion about the findings presented in the revised main paper.
>
> (2) **“Not very clear when different loss functions are used (equation 5 and equation 3)”**
> **“From the presentation it is not clear when the adversarial loss defined in equation 5…Can the authors provide some more details?”**
>
> Equation (5) is used for training the searched neural network by adversarial training using the loss of TRADES ($L_{\textup{TRADES }}$) and the Confidence-aware Adversarial Consistency (CAC) loss $L_{\textup{CAC}}$ define in Equation (4). Table 1 has two parts, the part titled “Without Adversarial Training” reports results of RNAC-CL models trained using Equation (3),  and the part titled  “With Adversarial Training” reports results of RNAC-CL models trained using Equation (5).
>
>
> (3) **“The paper contains few typos... Such as “for for”...”**
>
> We will scrutinize this paper carefully and fix all the typos and grammatical errors in the final version of this paper.
>
> (4) **“In the related work section, the authors do not mention the set of NAS approaches… such as [1,2].”**
>
> We agree with the reviewer, and we will add the discussion about such one-shot frameworks you mentioned to the final version of this paper.
>
> (5) **“How is equation 1 derived...”**
>
> We aim to find a tutor (teacher layer) for each student layer and hope that the attention maps of the student layer and the corresponding teacher layer are similar to each other. Therefore, we need to compute the loss between the attention maps of each student layer and each teacher layer. Inspired by [1], we calculate the distance between attention maps of student layer $i$ and a layer $j$ in the teacher model as $||\mathcal F(A_{s}^{i})/||\mathcal F(A_{s}^{i})\|| - \mathcal F(A_{t}^{j})/||\mathcal F(A_{t}^{j})\||||$.
> Each distance between a pair of student and teacher layers is weighted by a Gumbel weight. We optimize these Gumbel weights $g_{ij}$ during the search stage. For each student layer $i$, the layer $j$ in the teacher model with the maximum Gumbel weight $g_{ij}$ is selected as the teacher layer for the student layer $i$.
>
> (6) **“Figure 2a is not introduced…”**
>
> We will add the reference to Figure 2(a) in Section 3.2 of the revised paper.
>
> (7) **“The search and train loss functions are identical except for the latency variable...”**
>
> The search loss and the training loss are partially inspired by FBNet [2] and FBNet-V2 [3]. Similar to FBNet [2-3], our search loss and training loss include the cross-entropy term. Both search and training loss functions also include the attention loss ($L_{\textup{Attn}}$) and the KL-divergence $L_{\textup{KL}}$. However, different from the training loss, the search loss defined in Eq. (5) includes two different terms: the loss of TRADES ($L_{\textup{TRADES }}$) for adversarial training and the Confidence-aware Adversarial Consistency (CAC) loss $L_{\textup{CAC}}$ define in Equation (4). The searched neural architecture in the search stage is trained either using the training loss in Eq. (3) without adversarial training, or Eq. (5) with adversarial training. Such details will be added to the final version of this paper. Thank you for the suggestion, we will define the common terms in the search loss and the training loss in the final version of the paper.
>
>
>
> (8) **“All section references at the beginning of section 4 point to the appendix...”** **“The results are not well presented...”**
>
> The section references in Section 4.2 point to the appendix. We will follow your suggestions and revise the paper accordingly. In particular, we will shorten Table 1 and organize Section 3 and Section 4 of this paper so that we have more space to move the results in the appendix to the main paper. This will enable us to provide more detailed discussion about the findings presented in the main paper. Section D.1 in the appendix discusses the results in Table 1, which will also be moved to the revised main paper.
>
> (9) **“How did the authors select the values for the learning rate ... how these hyper-parameters are set.”**
>
> Our hyperparameters such as learning rate, momentum, batch size, and epochs, are based on [4] for the CIFAR-10 dataset. For the ImageNet dataset, our hyperparameter settings are based on FBNet-V2 [3]. The values of hyperparameters $\lambda_s$ and $\lambda_t$ in equations (2) and (3) are chosen from a candidate set ${0.01, 0.1, 0.1, 1.0, 10, 100}$ and are set to 1.0 for all experiments by standard cross-validation.

---

### Official Review · Reviewer_cjoR · 2024-03-24

**Q2-1 Originality-Novelty:** 2
**Q2-2 Correctness-Technical Quality:** 2
**Q2-5 Clarity Of Writing:** 3

**Q1 Summary And Contributions:**

The work proposes a novel approach to search for for a neural architecture that optimizes the tradeoff between robustness and prediction accuracy in a differentiable manner.

**Q2-3 Extent To Which Claims Are Supported By Evidence:**

3: Good: the main claims are supported by convincing evidence (in the form of adequate experimental evaluation, proofs, (pseudo-)code, references, assumptions).

**Q2-4 Reproducibility:**

3: Good: key resources (e.g. proofs, code, data) are available and key details (e.g. proofs, experimental setup) are sufficiently well-described for competent researchers to confidently reproduce the main results.

**Q3 Main Strengths:**

- Clear organization
- Precise formalization and convincing experimental analysis

**Q4 Main Weakness:**

I do not have major concerns.

**Q5 Detailed Comments To The Authors:**

In Section 4.2, it would be relevant to report a high-level discussion of the findings reported in the supplementary material.

**Q9 Complying With Reviewing Instructions:**

Yes

---

> ### Author Rebuttal · Authors · 2024-04-09
>
> We appreciate the review and the suggestions in this review. The raised issues are addressed below.
>
>
> **“In Section 4.2, it would be relevant to report a high-level discussion of the findings reported in the supplementary material.”**
>
> We will shorten Table 1 and reorganize Section 3 and Section 4 of this paper so that we have more space to move the results in the appendix to the main paper. This will enable us to provide more detailed discussion about the findings presented in the main paper. Following your suggestion, we will also add to Section 4.2 of the main paper a high-level discussion of the findings in the supplementary.

---

### Meta-Review · Area_Chair_Jd8T · 2024-04-24

This paper proposes distilling the knowledge of a robust teacher network into a student network. To inherit the teacher network's robustness properties, neural architecture search (NAS) is integrated to search for and connect the student network's layers to informative layers of the teacher network. Other methods, including TRADES and FastAT, are also integrated to enhance the student network's robustness. Overall, the idea looks novel, although it simply integrates several existing techniques in NAS, robustness, and knowledge installation. Besides all the reviewers' comments and concerns, it's not clear how much of the robustness comes from the architecture of the student network (through NAS) as the paper highlights, and how much comes from the integration of robustifying techniques